# Relationships Between Short-Term Exposure to an Indoor Environment and Dry Eye (DE) Symptoms

**DOI:** 10.3390/jcm9051316

**Published:** 2020-05-02

**Authors:** Maria A. Idarraga, Juan S. Guerrero, Samantha G. Mosle, Frank Miralles, Anat Galor, Naresh Kumar

**Affiliations:** 1Bascom Palmer Eye Institute, University of Miami, Miami, FL 33136, USA; mai47@miami.edu (M.A.I.); juansegr11@hotmail.com (J.S.G.); 2Environmental Health Division, University of Miami, Miami, FL 33136, USA; sgm71@miami.edu (S.G.M.); fmiralles@med.miami.edu (F.M.); NKumar@med.miami.edu (N.K.); 3Department of Ophthalmology, Miami Veterans Administration Medical Center, Miami, FL 33125, USA

**Keywords:** Dry Eye, Sick Building Syndrome, Bioaerosols, Indoor Air Pollution, Short-term Exposure, Older Buildings

## Abstract

Air composition influences Dry Eye (DE) symptoms as demonstrated by studies that have linked the outdoor environment to DE. However, there is insufficient data on the effect of short-term exposure to indoor environments on DE symptoms. We conducted a prospective experimental research, in which an older building served as an experimental site, and a newer building served as the control site. Indoor air quality was monitored in both buildings. One-hundred-and-ninety-four randomly selected individuals were interviewed in the afternoon exiting the buildings and de-identified responses were recorded. Self-reported DE symptoms were modeled with respect to experimental and control buildings, adjusting for potential confounders. The experimental site had 2-fold higher concentration of airborne particulate matter (24,436 vs. 12,213 ≥ 0.5 µm/ft^3^) and microbial colonies (1066 vs. 400/m^3^), as compared to the control building. DE symptoms were reported by 37.5% of individuals exiting the experimental and 28.4% exiting the control building. In the univariate analysis, subjects exiting the experimental building were 2.21× more likely to report worsening of DE symptoms since morning compared to the control building (*p* < 0.05). When adjusting for confounders, including a history of eye allergy, subjects from the experimental building were 13.3× more likely to report worsening of their DE symptoms (*p* < 0.05). Our findings suggest that short-term exposure to adverse indoor environmental conditions, specifically air pollution and bioaerosols, has an acutely negative impact on DE symptoms.

## 1. Introduction

Dry eye (DE) disease is a “multifactorial disease” of the ocular surface characterized by a loss of homeostasis of the tear film, and accompanied by ocular symptoms, in which tear film instability and hyperosmolarity, ocular surface inflammation and damage, and neurosensory abnormalities play etiological roles [1]. Inherent in this definition is the complex nature and multiple contributors of DE. Epidemiological studies using various definitions for DE have reported prevalence estimates between 5% and 50%, with higher frequencies seen in women and with increasing age [2]. Cost-of-illness analyses have estimated the burden of DE for the US healthcare system at approximately $3.84 billion, including indirect costs (e.g., loss of working hours) [3]. Along the many risk factors for DE, environmental conditions have been implicated in the onset and persistence of the disease [4]. Given that our eyes are directly exposed to air, there is biological plausibility that the properties and composition of air may alter the pre-corneal tear film and impact corneal nerve function [5]. Most of the epidemiological research in this regard has focused on the relationship between outdoor environment and DE, with air pollution being the metric most commonly linked to disease [4,6,7].

Less is known about the effects of the indoor environment, which encompasses a combination of air pollutants, relative humidity, air flow, and temperature, on ocular surface health. This is despite the fact that we spend most of our time indoors, and thus our indoor environment is critically important to our productivity [8]. Studies have demonstrated that short- and long-term exposure to adverse indoor environments, such as poor air quality, physical discomfort, noise, vibration, and overcrowding, pose health threats [9]. People living and/or working in buildings with funny odors, poorly regulated heating and air conditioning, dampness, elevated levels of air pollutants, and bioaerosols report a myriad of symptoms and signs that are placed under the heading of Sick Building Syndrome (SBS) or Building Related Illnesses (BRI) [10,11,12,13]. Although the term SBS is poorly defined [14], commonly reported SBS symptoms include headache, fatigue, shortness of breath, sinus congestion, cough, sneezing, eye, nose, throat and skin irritation, dizziness, and nausea [8,15,16,17,18].

In relation to the eye, one study surveyed 877 occupants in 12 public office buildings and found that 35% were dissatisfied with their indoor air quality, which correlated with reports of ocular discomfort (29%) [19]. Furthermore, ocular symptoms were found to be temporally related to the indoor environment. Another study measured formaldehyde levels in 100 homes and found that 20% had elevated indoor concentrations (≥0.81 ppm). Self-reported eye irritation (68%) and burning (60%), measured via yes/no, were common complaints in individuals living in the studied homes (*n* = 256) and 89% reported that symptoms resolved when leaving home [20]. Finally, perceived air quality was found to be related to objective markers of ocular surface health. In one study of university staff working in 100-year-old buildings (*n* = 173), perceived indoor environment (“dry air” in particular) correlated with a more rapid tear break up time (TBUT) (odds ratio (OR) 0.76, 95% confidence interval (CI) 0.64–0.90) [21]. In fact, when 10 subjects were experimentally exposed to office dust for 3 h in a climate-controlled chamber, TBUT decreased significantly [22]. However, not all studies have found relationships between the indoor environment and eye symptoms. A study in Brazil reported that self-reported DE symptoms did not relate to exposure levels of particulate matter (PM) and volatile organic compounds (VOCs) in two buildings in Rio de Janeiro [16]. Likewise, a study from Taipei found no associations between indoor air pollution (carbon dioxide (CO_2_) and VOCs) and self-reported eye dryness and irritation [23].

More information is thus needed to better understand the relationships between indoor environments and DE symptoms. This is because concentrations of some pollutants can be 2 to 5 times higher indoors than outdoors [24] and Americans spend approximately 90 percent of their time indoors [25]. Furthermore, manipulating the indoor environment is a more cost-effective approach to managing DE symptoms than changing outdoor exposures or providing individual level medical care. While studies have described the adverse effects of chronic exposure to air pollutants on respiratory, cardiovascular, and ocular health [26], an understanding of the effects of short-term exposure to indoor environments in diseases is lacking, including in relation to DE symptoms. This study bridges this knowledge gap by evaluating the frequency and change in DE symptoms in response to short-term exposure to two different building environments. We hypothesize that short-term exposure to an adverse environment will be associated with worsening DE symptoms as compared with a healthier environment.

## 2. Materials and Methods

The study was IRB exempt given the use of de-identified data.

### 2.1. Study Design

We used a prospective experimental research design, in which an older building, i.e., Calder Library, served as an experimental building, and a newer LEED (Leadership in Energy and Environmental Design)-certified building, Clinical Research Building (CRB), served as the control building. All subjects exiting these buildings between 12 and 6 PM were asked for their voluntary participation in the study. Individuals were interviewed in the afternoon as we found this time period to be optimal in capturing individuals who spent several hours in the building. When we tested morning time intervals, many individuals reported spending < 1 h in the building. All responses were recorded using a tablet and transmitted to our server. The questionnaire used can be found at dryeye.miami.edu/de_env/de_study. A total of 194 randomly selected individuals, >18 years of age, participated in the study. Subjects were asked questions about their demographics (age and sex), home environment (year home built, carpets in the home, and temperature in the home), co-morbidities (smoking status, contact lens use, refractive surgery, and history of eye allergies), and time entering and exiting the building. Our main outcome measures focused on DE symptoms, assessed with two questions: (1) “Are you currently experiencing any dry eye symptoms, such as dryness, irritation, or burning sensation?” Responses were coded as 1 = yes; 0 = no. (2) “Have your dry eye symptoms changed since the morning?” Responses were coded as 1 = worsened; 0 = no change or improvement. Presence and worsening of DE symptoms were modeled with respect to experimental and control environments (i.e., old vs. new building), adjusting for potential confounders.

### 2.2. Environmental Monitoring

Indoor air quality was monitored inside the two buildings on 15 November 2019 for 45 min each (Calder Library 10:30 am and CRB 11:50 am), using a set-up of multiple instruments placed on a tripod 4 feet (1.2 meters) from the floor, including AEROCET, a handheld particle counter; PRECISE, a real-time portable sensor for air quality monitoring; a bioaerosol impactor with soy agar plate; and a sterile cellulose nitrate membrane filter (CNMF). The same instruments and set up were used for both locations after instrument calibration. Air flow for both the CNMF and bioaerosol impactor was 28.5 L/min. General purpose nutrient media was added onto CNMF and was placed in an incubator for 48 h at 5% CO_2_ and 37.5 °C along with the soy plate. Airborne PM, bioaerosols, reactive gases, humidity, and temperature were all measured. The outside temperature during the time of monitoring was 82–84 °F (27–28 °C).

### 2.3. Statistical Analysis

Descriptive analyses were conducted to assess patient demographics, comorbidities, and DE measures. Student’s *t* tests and χ^2^ analyses were conducted to compare demographics and comorbidities between the populations exiting the two buildings. Odds ratios were computed to examine DE symptom presence and worsening by patient and DE measures. Multivariable analyses were then performed to examine differences in the main outcomes across experimental and control conditions adjusting for potential confounders, including demographics and interactions between variables of interest. *p*-values of ≤ 0.05 were considered significant. All analyses were conducted in STATA Ver 14.2 [27].

## 3. Results

### 3.1. Study Population

Out of the 194 subjects who participated in the study, 190 subjects completed the survey (102 in CRB and 88 in Calder Library). Overall, 56.2% of the subjects were female and most of them were less than 30 years of age (61.6%). Individuals exiting the control building were older (mean 35.7 standard deviation (SD) 13.1 years vs. 31.2 SD 14.0 years) and less likely to be female (54.8% vs. 59.8%) than the case building. In addition, individuals spent more time in the control building than the experimental building (mean 3.4 SD 2.9 h vs. 2.3 SD 2.2 h). Comorbidities and home information were similar between the two groups (Table 1).

### 3.2. Indoor Air Quality

The experimental building had 2× higher concentration of airborne PM (24,436 vs. 12,213 number of particles ≥ 0.5 µm/feet^3^) and microbial colonies (1066 vs. 400 number of airborne microbial colonies/m^3^), as compared with the control building (Figure 1). Relative humidity (RH) and carbon monoxide (CO) were also higher in the experiment building as compared with the control building. However, temperature was slightly higher in the control building, and nitrogen dioxide (NO_2_) was 5 ppm higher in the control building compared with the experimental building (Table 2).

### 3.3. Dry Eye Symptoms

DE symptoms, including sensations of dryness, irritation, and burning, were reported by 37.5% of individuals exiting the Calder Library (experimental building) and 28.4% individuals exiting CRB (control building). In a similar manner, 29.1% vs. 15.4% reported that their DE symptoms worsened since morning, respectively (Figure 2). In the univariate analysis, the difference in presence of DE symptoms were not statistically different across two buildings (OR = 1.51; 95% CI = 0.82–2.78; *p* = 0.19) (Table 3). However, after adjusting for demographics, comorbidities, home environment, and time spent in the building, subjects from the experimental building were 3.89 more likely to report DE symptoms (OR =3.89; 95% CI = 1.21–12.5; *p* < 0.05) than those from the control building. Table 4 displays the results of several multivariable models that considered the effects of various independent variables on DE symptom presence. The full model (right most column) considered building data, eye and systemic co-morbidities, demographics, and home information. Subjects with a history of eye allergy were 10 times more likely to report DE symptoms while in the experimental building compared to the control building (OR = 10.03; 95% CI = 3.13–32.18; *p* < 0.01).

Even in the univariate analysis, worsening of DE symptoms since morning varied across subjects exiting the experimental and control buildings (OR = 2.21; 95% CI = 1.08–4.55) (Table 5, left most column). When adjusted for demographics and comorbidities, including a history of eye allergy, eye surgery, and time spent in the building, subjects from the experimental building were 13.30 times more likely to report worsening of their DE symptoms (OR = 13.30, 95% CI = 2.80–63.31; *p* < 0.01) as compared to the control building (Table 5, right most column). Other factors that impacted DE symptom worsening included a self-reported history of ocular allergy (OR = 4.05, 95% CI = 1.18–13.92, *p* < 0.05) and a history of LASIK (OR = 9.74, 95 CI% = 2.91–32.58, *p* < 0.01).

### 3.4. Time in Building and its Impact on DE Symptoms

Interestingly, regarding the univariable analysis, people who spent a shorter time in the building (<1 h) were more likely to report DE symptoms worsening compared to individuals who spent 1–3 h and >3 h. However, when adjusting for other variables, including building and time interaction, there was a 1% increase in the odds of reporting worsening of DE symptoms per hour of staying in the Calder building (OR = 1.01; 95% CI=1.00–1.02; *p* < 0.05) as compared to the CRB (Table 5).

## 4. Discussion

This study investigated the impact of short-term exposure to indoor air environment on ocular surface symptoms. We found that individuals exiting buildings with higher levels of air pollutants and microbial loads were more likely to report worsening of DE symptoms than those exiting buildings with lower loads. Our findings are in general agreement with the literature as elevated levels of air pollutants, microbial counts, and humidity have been previously associated with adverse health outcomes, including asthma, allergies, and immunological disorders [13,28]. In our study, we found a 2.2-fold increased risk in DE symptom worsening in individuals exiting the Calder Library vs. the CRB. The risk increased to 13-fold when adjusting for other co-morbidities, such as history of LASIK and eye allergies. In a similar manner, individuals in Copenhagen working in buildings associated with SBS had a 1.5 times increased frequency of eye complaints (tiredness, itching, irritation, and dryness) compared to the general population (42% vs. 27%) [29].

Short-term exposures to other pollutants have also been associated with DE symptoms and signs. For example, in a Korean study, 33 individuals with evaporative DE (TBUT ≤5 s) underwent two eye exams, 2-month apart. Simultaneously, mean ground-level ozone (O_3_) concentrations for the week prior to the clinic visits were obtained via monitoring stations near subjects’ homes. Changes in O_3_ levels correlated with changes in DE parameters, including OSDI scores (*r* = 0.30, *p* = 0.0006) and tear secretion (*r* = −0.36, *p* = 0.001) [30].

It is interesting that in our present study, while air particulate levels and microbial counts were higher in the older building, gaseous pollutants (NO_2_ in particular) were higher in the newer building. This finding may be explained by examining the proximity of a building to other sources of pollution. For example, the CRB is next to an expressway, and intrusion of outdoor NO_2_ from automobiles can be a potential explanation for the elevated levels as compared to the Calder Library which is three blocks away from the expressway. In fact, VOCs, and specifically NO_2_, have also been linked to DE. A case-crossover study in Taiwan examined relationships between 25,818 subjects with a diagnosis of DE (via International Classification of Disease codes) and outdoor air pollutants, temperature and RH (extracted from environmental monitoring stations). Subjects served as their own controls as exposure on the day of the first DED diagnosis was compared to average exposures on four different days, one and two weeks before and after the date of DED. Multivariate analyses found significant association between CO (OR = 1.116; 95% CI = 1.026–1.214; *p* < 0.01), NO_2_ (OR = 1.068; 95% CI = 1.037–1.100; *p* < 0.001), RH (OR = 0.930; 95% CI = 0.910–0.949; *p* < 0.001), and temperature (OR = 1.010; 95% CI = 1.005–1.015; *p* < 0.001) with DED occurrence [31]. In other words, a 10-ppb increase in NO_2_ was associated with a 6.8–7.5% increase in DED occurrence. Our findings highlight that different buildings can have different sources/types of air pollutants, which can have different effects on human health.

As above, one factor to consider when evaluating sources of air pollution is the heating, ventilation, and air conditioning (HVAC) design. For example, building materials may become a source of air pollution, such as VOCs from paint, sealants from wood and plyboards, and microbes from mortar. These pollutants are released for many weeks to months after construction is complete [32]. Buildings with poor ventilation may impede the dispersion of VOCs. Thus, VOCs concentration can increase through new emission and recycling of the VOCs already trapped inside the building. This factor is often affected by year of building construction. For example, buildings constructed after 1970 (such as the CRB) often utilize energy conservation methods including improved insulation, reduced air exchange, increasing air conditioning, and construction without opening windows [33]. While heating and cooling in such building is cost-effective, if pollution sources enter the building, such as gaseous pollutants from car emissions, lack of air exchange can cause a build-up of pollutants that can be hazardous to occupants’ health.

There is biologic plausibility on why air pollutants and microbial contamination may cause worsening of DE symptoms as both have been linked to oxidative stress and inflammation. In animal studies, topical administration of PM_2.5_ on mouse corneas led to ocular surface damage similar to that seen in dry eye in humans [34,35,36]. Specifically, after 14 days of exposure, the PM_2.5_-treated group had decreased TBUTs (~2.5 s vs. ~6.5, *p* < 0.05) and decreased tear volumes (~1.5 mm vs. ~6 mm, *p* < 0.05) compared with the saline-control group. At the same time, inflammation markers on the ocular surface (tumor necrosis factor-α (TNF-α) and NF-κB) were elevated in the PM_2.5_ versus control group [35]. Humidity is an interesting risk factor as both low and high humidity have been associated with DE [37,38]. This is likely because low humidity can result in aqueous loss, whereas high humidity favors the growth, transmission, and survival of airborne microorganisms [38,39,40], thus an optimal middle range is between 45% and 50%, as recommended by the Environmental Protection Agency (EPA).

As with all studies, the findings of this research must be interpreted in light of the study limitations. First, we included individuals exiting two buildings at a specific date range and time frame. Second, we lack information on the location and activities undertaken within the building, which is important as exposure to air pollution varies within a building. For example, basements often have pollutants with higher molar mass, whereas higher floors often have pollutants with lighter mass which are more likely to float, such as dust and VOCs, including NO_2_ and O_3_. Likewise, areas facing sun and/or shade will have different exposure to heat, humidity and pollutants than those located in the center. Furthermore, actions within buildings, such as continuous use of computer screens and time spent reading books, specifically relevant in Calder Library, can impact symptoms of SBS and eye symptoms [41,42,43,44,45,46]. Future studies will thus need to incorporate information from multiple buildings and consider subject location (basement vs. upper floors) and activity (computer use) within a building. Furthermore, the building characteristics themselves are not identical, with CRB having a glass façade, which likely contributes to varied environmental profiles. Third, we measured select indoor air pollutants while others, such as 2-Ethyl-1-hexanol (2EH), were not directly measured [17]. However, we chose to measure pollutants that have been most closely linked to DE. Fourth, DE symptoms were collected by self-report and no data is available on DE signs or medication history (e.g., antihistamines for eye allergies). It is known, however, that DE symptoms and signs are often discordant [47], and we focused on DE symptoms as they are the main drivers of disease morbidity. Finally, generalizability to other populations is limited based on the large sample of young participants.

Despite these limitations, the findings of this research are important as they highlight that management of the indoor environment may be one strategy for protecting occupational health. While our findings are not enough to classify Calder library as an SBS building, we recommend that certain steps be taken to improve building environments. First, the monitoring of air pollutants and bioaerosols is needed to understand the air quality profile in a particular building, as sources of pollution differ by building. Second, building specific remediation plans should be developed to improve indoor air quality by strategies such as removing carpeting and controlling humidity and ventilation [12,13,48,49,50,51]. In buildings with high levels of VOC, indoor plants, such as *O. microdasys* [52] can be installed to remove gaseous pollutants. In buildings with high levels of air particulates, photocatalytic air purifier can be run during unoccupied building hours [53]. In-room air cleaners with high-efficiency particulate air (HEPA) or combined HEPA and ultraviolet germicidal irradiation (UVGI) air cleaning technology can also be used to remove airborne pathogens [54,55,56]. Finally, individuals spending time in buildings need to take proactive measures, such as wearing protective eyeglasses, or using artificial tears and antihistamines, based on individual susceptibility. These measures are important as symptoms of SBS, including ocular symptoms, have a negative impact on quality of life [57] and lead to anxiety, decreased productivity, spent resources investigating complaints, and avoidance behaviors [58,59]. These negative effects furthermore support the importance of improving air quality indoors. Further research is needed to evaluate whether approaches outlined above can modulate the indoor environment and be beneficial to ocular health.

## Figures and Tables

**Figure 1 jcm-09-01316-f001:**
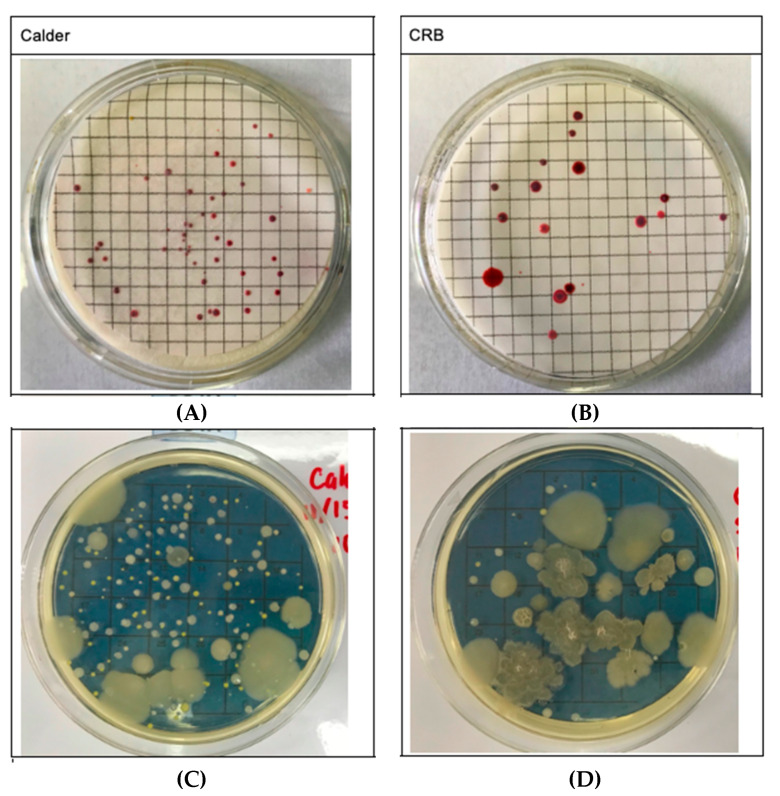
Microbial growth of air samples collected on the sterile gridded cellulose nitrate membrane filter and cultured using the general-purpose nutrient media purchased from MilliporeSigma™ MHA00P2TT (**A**,**B**). Soya Agar plate placed in a bioaerosol impactor (**C**,**D**). Both samples in each building were collected simultaneously (**A**,**C** in Calder) and (**B**,**D** in CRB) at a flow rate of 28.5 L/m and incubated for 48 h at 5% CO_2_ and 37.5 °C. Number of airborne microbial colonies/m^3^ was quantified by counting colonies on soya plates (**C**,**D**). Note that color differences between colonies qualitatively highlight that different species of microbes (bacterial and fungal) are present in the samples.

**Figure 2 jcm-09-01316-f002:**
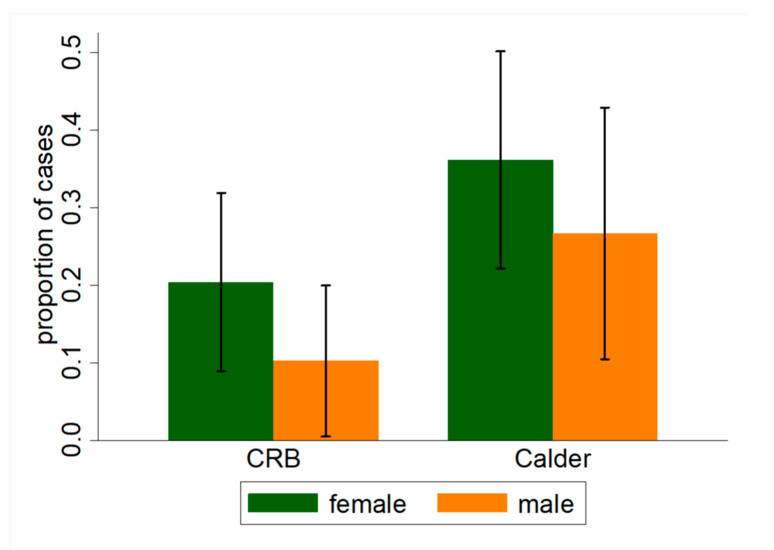
Proportion of individuals who reported worsening Dry Eye (DE) symptoms grouped by building (CRB = Clinical Research Building, control versus Calder Library, experimental) and sex.

**Table 1 jcm-09-01316-t001:** Population sample composition by building.

	CRB (Control)	Calder (Experimental)	*p*-Value
**Demographics**			
Age, mean (SD)	35.7 (13.1)	31.2 (14.0)	0.02
Female, % (*n*)	54.8% (57)	59.8% (52)	0.05
**Comorbidities**			
LASIK, % (*n*)	13.2% (14)	21.6% (19)	0.12
Eye allergy, % (*n*)	18.7% (20)	26.5% (15)	0.69
Current contact lens use, % (*n*)	17.9% (19)	22.7% (20)	0.53
Active smoker, % (*n*)	8.6% (9)	10.2% (9)	0.46
Eczema, % (*n*)	3.7% (4)	4.4% (4)	0.82
**Building**			
Hours in building, mean (SD)	3.4 (2.9)	2.3 (2.2)	0.005
**Home environment**			
Subjects living in home built before 1980, % (*n*)	20.2% (19)	19.1% (14)	0.87
Carpets at home, % (*n*)	39.4% (41)	40% (34)	0.94

CRB = Clinical Research Building; *n* = number in group; LASIK = laser-assisted in situ keratomileusis; CI = Confidence interval; *n* = number in each group; SD = standard deviation.

**Table 2 jcm-09-01316-t002:** Indoor environmental metrics inside Calder Library and CRB.

Variable, Mean (CI)	Calder	CRB	Difference
Number of particles ≥ 0.5 µm/feet^3^	24,436 (23,338–25,533; 45)	12,213 (11623.30–12802.92; 45)	12,223
Number of particles ≥ 5 µm/feet^3^	1372 (1231–1512; 45)	799 (683.19–914.14; 45)	573
Relative humidity (%)	62.00 (61.71–62.29; 45)	59.54 (59.23–59.84; 45)	2.46
CO(Carbon monoxide (ppm))	12.87 (11.18–14.55; 122)	11.41 (10.22–12.61; 88)	1.45
NO_2_ (Nitrogen dioxide (ppm))	7.70 (6.68–8.73; 122)	12.85 (12.18–13.53; 88)	−5.15
Temperature	19.03 (18.70–19.37; 122)	20.81 (20.50–21.12; 88)	−1.78
Number of airborne microbial colonies/m^3^	1066	400	666

CI = confidence interval; ppm = Parts Per Million.

**Table 3 jcm-09-01316-t003:** Odds ratio of the presence and worsening of DE symptoms with respect to selected covariates.

Selected Variables	Presence of DE Symptoms (1 = Yes; 0 = No) OR (95% CI, *p*-Value)	Worsening of DE Symptoms (1 = Yes; 0 = No) OR (95% CI, *p*-Value)
**Building Location**		
CRB (reference)	1	1
Calder	1.51 (0.82–2.78; 0.19)	2.21 (1.06–4.60; 0.03)
**Time Spent Inside the Building (hours)**		
<1 h (reference)	1	1
1 to 3 h	0.54 (0.26–1.14; 0.10)	0.72 (0.30–1.72; 0.45)
>3 h	0.34 (0.15–0.76; 0.01)	0.45 (0.18–1.16; 0.09)
**Contact Lens Wear**		
No (reference)	1	1
Not wearing now	0.48 (0.13–1.81; 0.27)	1.83 (0.58–5.84; 0.29)
Yes	1.53 (0.73–3.21; 0.25)	2.20 (0.94–5.13; 0.06)
**Smoking**		
No (reference)	1	1
Inactive (smoked in the past)	0.76 (0.29–1.95; 0.57)	0.43 (0.12–1.56; 0.19)
Yes	0.45 (0.12–1.68; 0.22)	0.72 (0.19–2.72; 0.63)
**LASIK Surgery**		
No (reference)	1	1
Yes	4.39 (1.96–9.78; <0.001)	5.67 (2.26–14.25; <0.001)
**Self-reported Eye Allergies**		
No (reference)	1	1
Yes	6.84 (2.75–17.00; 0.00)	4.37(1.80–10.61; 0.00)
**Eczema**		
No (reference)	1	1
Yes	1.28 (0.29–5.55; 0.74)	1.30 (0.24–7.02; 0.76)
**Home Construction Year**		
Before 1980 (reference)	1	1
In or after 1980	0.86 (0.39–1.91; 0.69)	1.45 (0.53–3.90; 0.32)
**Temperature Indoor (°F)**		
<70 °F (<21 °C) (reference)	1	1
70–75 °F (21–23 °C)	1.15 (0.57–2.33; 0.69)	1.14 (0.52–2.53; 0.75)
>75 °F (>23 °C)	0.22 (0.06–0.90; 0.02)	0.12 (0.01–1.05; 0.02)
**Carpet in the House**		
No (reference)	1	1
Yes	1.84 (0.98–3.46; 0.05)	1.98 (0.95–4.16; 0.06)

OR = odds ratio; CI = confidence interval; LASIK = laser-assisted in situ keratomileusis.

**Table 4 jcm-09-01316-t004:** Multivariable analyses of variables associated with the presence of DE symptoms at the time of interview.

Variables	Dry Eye Symptoms at Present (0 = No; 1 = Yes)					
Building (0 = CRB, 1 = Calder Library)	1.51			3.81 **	4.01 **	3.89 *
	(0.82–2.78)			(1.42–10.23)	(1.40–11.48)	(1.21–12.50)
Time Spent Inside the Building (hours)		0.92		0.44 *	0.43 *	0.51 *
		(0.80–1.05)		(0.22–0.86)	(0.20–0.90)	(0.27–0.97)
Building × Time Spent in the Building (hours)			1	1.007 *	1.007 *	1.01 *
			(0.1–1.00)	(1.001–1.013)	(1.001–1.014)	(1.00–1.01)
Eye Allergy Status (0 = no, 1 = yes)					5.97 **	10.03 **
					(2.41–14.79)	(3.13–32.18)
Eczema Status (0 = no, 0 = yes)						0.92
						(0.14–6.25)
Age (years)						1.01
						(0.98–1.05)
Sex (0 = female, 1 = male)						0.76
						(0.34–1.70)
LASIK Surgery (0 = no, 1 = yes)						3.74 **
						(1.41–9.92)
Contact Lens Wear (0 = no; 1 = not now, 2 = yes)						0.9
						(0.53–1.54)
Smoking Status (0 = no; 1 = previous; 2 = yes)						0.32 *
						(0.11–0.87)
Home Construction Year (0 = before 1980; 1 = 1980 and after)						1.07
						(0.34–3.35)
Temperature Indoor (°F)						0.92
						(0.81–1.03)
Carpet in the House (0 = no, 1 = yes)						1.58
						(0.70–3.57)
Observations	190	195	187	187	187	169

Robust 95% confidence intervals in parentheses; ** *p* < 0.01, * *p* < 0.05.

**Table 5 jcm-09-01316-t005:** Multivariable analysis of factors associated with worsening DE symptoms after spending time in the selected buildings.

Variables	Worsening of DE Symptoms = 1, Otherwise = 0 Odds Ratio (95% Confidence Interval)					
Building (0 = CRB, 1 = Calder Library)	2.21 **			6.95 ***	6.97 ***	13.30 ***
	(1.08–4.55)			(2.09–23.04)	(2.03–23.91)	(2.80–63.31)
Time Spent Inside the Building (hours)		0.95		0.45 ***	0.45 ***	0.41 **
		(0.82–1.10)		(0.26–0.78)	(0.25–0.79)	(0.19–0.92)
Building × Time Spent in the Building (hours)			1	1.01 ***	1.01 ***	1.01 **
			(0.1–1.00)	(1.00–1.01)	(1.00–1.01)	(1.00–1.02)
Eye Allergy Status (0 = no, 1 = yes)					3.35 **	4.05 **
					(1.31–8.57)	(1.18–13.92)
Eczema Status (0 = no, 0 = yes)						2.33
						(0.28–19.35)
Age (years)						0.99
						(0.94–1.04)
Sex (0 = female, 1 = male)						1.19
						(0.44–3.23)
LASIK Surgery (0 = no, 1 = yes)						9.74***
						(2.91–32.58)
Contact Lens Wear (0 = no; 1 = not now, 2 = yes)						1.70 *
						(0.92–3.11)
Smoking Status (0 = no; 1 = previous; 2 = yes)						0.27 *
						(0.07–1.02)
Home Construction Year (0 = before 1980; 1 = 1980 and after)						1.2
						(0.30–4.80)
Temperature Indoor (°F)						0.88
						(0.75–1.03)
Carpet in the House (0 = no, 1 = yes)						1.73
						(0.66–4.56)
Observations	169	169	166	166	166	152

Robust 95% confidence intervals in parentheses; *** *p* < 0.01, ** *p* < 0.05, * *p* < 0.1.

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
