# Peer review of "Relationships Between Short-Term Exposure to an Indoor Environment and Dry Eye (DE) Symptoms"

_jcm, 2020, doi:10.3390/jcm9051316_

Round 1

Reviewer 1 Report

Indoor environment and dry eye is a combination by pollutants, relative humidity, air flow and temperature. Especially when looking for the sensation dryness and irritation ( 57). In the introduction the reaction towards the assumption of the newer building vs de older one towards the differents of "quality of the indoor environment" is very random. The relative humidity in and outside, air pollution in and outside / different countries could have been taken into account. 

Methodes: at what hight was the portable sensor setup?  Was the same instrument used, calibrated? Time frame?  Measurement at the same day?  winter?summer? Sunny day? The outside temperature plays part in temperature, airflow and relative humidity inside a building. Also the activities are important, more people walking around gives a other air flow than a static office environment, Are the measurement taken in a library with books? More dust? 

98 allergies? or eye allergies as noted in the table?  Given the use of deidentified data ... 

table 4 is not helping the reader to understand the situation here. 

The generalization (204) is that true? When is a building a SBS building? Reflection needed on the use of the two building, not only explaining the highway . Measurements taken: desk hight?  eye hight? Relative Humidity? (253) wha tis the optimal relative humidity for the eyes? Cleaning of the buildings?  No controle of the work the participants were doing? Measurement hight ! Bias is also there with the questions taken right? How was the communication for the participations, how  tare they recruited, bias  there?  why 12 to 6 PM, bias there in remembering the comfort of the eyes in the morning? How were they selected, temperature at home? How was that questioned?  

Figure 1: collected simultaneously, after45 min, at what time a day? Looking at the figure and the explanation doesn't help.

Material and Methods is not clear to me: I cannot reproduce this investigation. Clarification needed.

Dust and particles are more likely to float in a hard floor.

Assuming, the participants were not using medications such als anti histamine?   

Reviewer 2 Report

The purpose of this article was to determine whether short-term exposure to an older, as compared to younger, building is associated with an increase in dry eye symptoms. The authors found that short-term exposure to adverse indoor environmental conditions, as in the older building, have an acutely negative impact on dry eye symptoms. This article is extremely well written; indeed, it is the best written of all the many papers I have reviewed for medical journals this year. This reviewer has only a few recommendations.

  • Add a y-axis title to the Figure 2 graph.
  • Explain in Tables 4 and 5 the differences between the six vertical columns that contain data. As currently presented, it is not clear why there are 6 columns (i.e. instead of one).
  • In lines 96 and 169, and in Tables 4 and 5, replace the word "gender" with "sex." Gender refers to behavioral differences, whereas "sex" refers to biological differences (i.e. female versus male) (Wizemann TM, Pardue M-L, Eds., and the Committee on Understanding the Biology of Sex and Gender Differences, Institute of Medicine. Exploring the Biological Contributions to Human Health. Does sex matter? National Academy Press, Washington, DC, 2003, pp. 1-287).
  • A limitation to this study is that the authors did not account for the time reading books or text in the library. Such reading may have been a contributing factor to the increased dry eye symptoms in the older building. Research has linked reading text and dry eye (e.g. Karakus S, Mathews PM, Agrawal D, Henrich C, Ramulu PY, Akpek EK. Impact of Dry Eye on Prolonged Reading. Optom Vis Sci. 2018 Dec;95(12):1105-111), and a recent study has shown that reading books, and not just staring at computers, also increases dry eye symptoms (Prabhasawat P, Pinitpuwadol W, Angsriprasert D, Chonpimai P, Saiman M. Tear film change and ocular symptoms after reading printed book and electronic book: a crossover study. Jpn J Ophthalmol. 2019 Mar;63(2):137-144). The authors should address this limitation in their Discussion section.

Round 2

Reviewer 1 Report

Dear Authors, 

The changes mades are sufficient to answer my questions. 

Best regards